# University students' perspectives, planned uptake, and hesitancy regarding the COVID-19 vaccine: A multi-methods study

**Madeleine Mant** [1]*, **Asal Aslemand**[2], **Andrew Prine**[3], **Alyson Jaagumägi Holland**[4]

**1** Department of Anthropology, University of Toronto Mississauga, Mississauga, Ontario, Canada,
**2** Department of Mathematical & Computational Sciences, University of Toronto Mississauga, Mississauga,
Ontario, Canada, **3** Groves Memorial Community Hospital, Fergus, Ontario, Canada, **4** Department of
Family Medicine, Faculty of Medicine, McMaster University, Hamilton, Ontario, Canada

* maddy.mant@utoronto.ca

journal.pone.0255447

University of the Health Sciences, UNITED STATES

**Data Availability Statement:** Data cannot be
shared publicly because interview and survey data
hold potentially identifiable sensitive information
regarding the student participants and it would

## Abstract

### Purpose

To investigate university students' willingness to receive a COVID-19 vaccine when it
becomes available to them.

### Method

A multi-methods approach was used—online convenience sample surveys and semi-structured interviews—of young adults attending a large Canadian public university. Two survey
samples were collected (June 20-July 28, 2020 and September 22-October 17, 2020).
Semi-structured interviews were conducted following each survey, interviewing 20 students
in each round.

### Results

In June 77.8% of surveyed students (n = 483) were willing to get the COVID-19 vaccine; in
September 79.6% were willing (n = 1269). Multinomial and binary logistic regression analyses found that increasing perception of the severity of COVID-19 predicted the likelihood
that a respondent was willing to get the COVID-19 vaccine in both surveys. In the latter survey students who indicated they would be encouraged to get the COVID-19 vaccine if their
doctor/pharmacist recommended it were 76 times more likely to be willing to get the vaccine
than those who would not be encouraged by medical advice. Interviews revealed concerns
about the speed of the vaccine roll out, safety, and efficacy.

### Conclusions

The majority of university students intend to get the COVID-19 vaccine, but there are
nuanced concerns about efficacy and safety that must be taken into account by public health
authorities as the vaccine becomes available to this group. Ensuring that family doctors,
pharmacists, and other front-line healthcare workers have consistent and clear information

therefore be unethical to make this public and would undermine the minimal risk ethical committee agreement and consent process. Participants did not provide consent for their transcripts and linked survey responses to be made public. Data are available from the University of Toronto Human Research & Ethics Unit (HREU) Research Oversight & Compliance Office (ROCO) (contact via ethics.review@utoronto.ca) for researchers who meet the criteria for access to confidential data.

**Funding:** MM: This study was funded by the University of Toronto COVID-19 Action Initiative. The funders had no role in study design, data collection and analysis, decision to publish, or preparation of the manuscript.

**Competing interests:** The authors have declared that no competing interests exist.

regarding the benefits of vaccination will be critical to encouraging uptake among young adults.

## Introduction

The SARS-CoV-2 (COVID-19) pandemic has resulted in over 2.5 million deaths and more than 116.1 million reported cases worldwide, as of March 7, 2021 [1]. Preventative efforts have been of supreme importance as caseloads rise globally, with vaccines the focus of international attention [2]. As of March 15, 2021, four vaccines have been approved for use in Canada, with 81 in clinical development and a further 182 in pre-clinical development [3]. Vaccine uptake by the public will be critical to quell the COVID-19 pandemic. Vaccine hesitancy, "including complacency and lack of confidence and convenience" was identified by the World Health Organization as a top ten threat to global health in 2019 [4]. The main objective of this research was to investigate predictors of university students' willingness to receive a COVID-19 vaccine once one was available. This research is part of a larger study investigating university student attitudes towards and perceptions of the COVID-19 pandemic.

University-aged students are a unique demographic group with underlying illness experiences and media and information consumption habits that differ from older adults. Previous investigations of university-aged students and vaccines have found low seasonal flu vaccine uptake among students [5–7] and low vaccine knowledge [6, 8]. Abalkhail et al.'s [9] study of medical students found low uptake of the seasonal flu vaccine, with the assumption that they were not at risk of catching influenza and potential vaccine side effects listed as the students' most common reasons for avoiding vaccination. Sandler and colleagues' [8] qualitative study concerning vaccine knowledge showed that parental influence played a large role in students' decision-making. Seanehia et al. [10] provided French university students with a series of hypothetical situations concerning vaccination and found that an epidemic situation correlated with the highest vaccine acceptance. While Schmid and colleagues [11] note that age is an inconsistent predictor of seasonal flu vaccine uptake, they also demonstrate that most research focuses upon the general public rather than a specific risk group. Young adults are often subsumed within the broader 'adult' category, which can include all individuals over the age of 18. University-aged adults are of specific interest as the second wave of COVID-19 in Canada saw rising numbers of young adults being infected and hospitalized compared to the first wave [12]. As a result, understanding young adult perceptions of COVID-19 and their intention to receive a vaccine is essential for public health planning.

The Health Belief Model (HBM) is a framework utilized for predicting health behaviours in relation to individuals' perceptions and beliefs. The model outlines how an individual's perception of the severity of the effect an illness may have on their life, the benefits of engaging with health behaviours, and barriers to engaging with health behaviours affect their actions [13, 14]. Three categories are investigated to predict an individual's engagement with a health behaviour: individual perceptions, modifying factors, and likelihood of action [14, 15]. The HBM has been used to study intentions of vaccine uptake in varied groups [16–18], and most recently has been applied to the COVID-19 vaccine [19, 20]. In this research, we investigated individuals' perceptions of the severity of COVID-19, modifying factors (i.e., demographic variables, previous uptake of seasonal flu vaccine), their likelihood of action (i.e., taking the COVID-19 vaccine when it became available to them), and stated reasons for taking or not taking the vaccine (perceived benefits and barriers).

Few studies have yet been published concerning university student perceptions of the COVID-19 vaccine [21, 22]. Faase and Newby [23], in an Australian survey, found that younger age (18–29) was associated with low engagement with health protective behaviours regarding COVID-19. Barello and colleagues [20, p.782] note that "students are a good target for educational campaigns as they are still in their training period and are open to changing their habits." As global vaccination campaigns are underway and Canada undertakes its largest ever mass vaccination campaign, it is necessary to have finely grained understandings of public intentions regarding uptake of the COVID-19 vaccine, particularly given current plans for phased vaccine rollouts that generally place university-aged students as a low priority group [24]. For educational campaigns to be effective and widespread vaccine uptake to be attained, it is necessary to understand the perceptions of this particular group regarding perceived severity, barriers, and benefits, to effectively frame public health cues to action.

## Methods

We employed quantitative and qualitative methods to obtain nuanced perspectives from university students regarding their likelihood of getting the COVID-19 vaccine. This mixed-methods approach involved an online questionnaire and semi-structured individual interviews. The online questionnaire was circulated in June/July 2020 and in September/October 2020. This project is part of a larger study exploring young adults' perceptions, health behaviours, and responses to COVID-19. Ethics approval was granted by the University of Toronto Research Ethics Board (#39169).

### Recruitment

The two online surveys were convenience samples of students at a large Canadian public university. Participants were recruited through social media, the Department of Anthropology's website, and through the authors sharing the survey link with student clubs and colleagues. Only students currently enrolled at the university were allowed to participate. Survey participants were provided with a Letter of Information including full details of the study objectives, data protection, and confidentiality. Participants did not have to answer every question and were offered compensation in the form of entry into a draw for a $50 CAD gift card; the draw was not linked to participants' survey responses. Consent was indicated by participants reading the Letter of Information on the opening page of the survey and clicking through to the second page of the survey and answering questions.

### Survey and interview design

The survey was adapted with permission from the authors of a previous survey concerning university students and the 2003 Severe Acute Respiratory Syndrome (SARS) outbreak, who piloted and validated the survey [25]. Our team adapted the language of the original questions to refer to COVID-19 and added further questions concerning the possibility of taking the vaccine. Questions concerning reasons for and against taking the vaccine were adapted from Ramsey and Marczinski's validated study concerning H1N1 vaccine uptake in college students [26]. No pilot testing of the new vaccine questions was undertaken.

### Survey questions

The survey opened with demographic questions including age, gender, program of study, place of residence, and household income. Participants in both surveys were asked: "If a vaccine for COVID-19 were to become available, would you want to get it?" Respondents could

choose to indicate their willingness to get the vaccine as "Yes", "No", or "Not sure/Undecided." If the respondent selected "Yes" or "No" they were directed to a free text box with the option to expand their answer.

In the June/July survey, there were seven predictor variables: age, gender, program of study, level of household income, whether they have been personally affected by COVID-19 through personal illness or the illness of a friend or family member, self-described anxiety concerning contracting COVID-19 after hearing a media report about the disease (7-point Likert scale, "Not fearful/anxious" to "Very fearful/anxious"), and perception of the severity of COVID-19 (7-point Likert scale, "Not severe" to "Very severe"). The September/October survey included three additional predictor variables, asking whether respondents usually get the seasonal flu vaccine, whether they got the seasonal flu vaccine in 2019, and whether they would be encouraged to get the COVID-19 vaccine if their doctor or pharmacist recommended it. Vaccine cost and government-mandated vaccine reception were not considered as factors in this research. While specific immunization schedules vary by province, routine vaccinations for babies, children, and adults are offered free of charge across Canada. During the second survey period, Canadian Prime Minister Justin Trudeau confirmed earlier promises that the vaccine would be free for all Canadians [27]. In Canada, vaccinations are not mandatory, though the provinces of Ontario and New Brunswick require proof of vaccination for children to attend school [28, 29]. Exemption clauses exist for medical or philosophical reasons.

## Data analysis

Basic frequencies were used to describe all variables. Age and perception of severity were treated as continuous variables, all remaining predictor variables were treated as categorical variables. Two levels of gender identification were used for the analyses (male, female) for which there was a sufficient percentage of complete cases, with females as the reference category. Self-described anxiety was grouped as low (1–3), moderate (4,5), and high (6,7). In the June/July survey we employed a multinominal logistic regression, since there were three possible responses measured for the dependent variable ("Yes", "No", "Not sure/Undecided"). In the September/October there were two levels of responses recorded for the dependent variable ("Yes", "No"), thus we employed a binary logistic regression analysis. Data analyses were conducted using R version 4.0.3 and p-values of <0.05 were considered statistically significant.

## Interview participant recruitment

At the end of the survey, participants were given the option to enter their email address if they were interested in taking part in an interview. We purposefully sought gender balance in the interviews, since survey participants were majority female. The interview participants were emailed a Letter of Information outlining the study objectives, their right to withdraw, and how their data would be stored. Written informed consent was gathered from each interview participant before the interview was conducted. Interviews were conducted over Skype or Zoom by MM and lasted approximately one hour. A semi-structured interview guide was used, following topics introduced in the survey. The interviews included the following question directly concerning vaccines: "If/when there is a vaccine available for COVID-19 would you get it?" Interview participants received a $20 CAD gift card for their time. The first set of interviews (n = 20) took place between July 18 and August 1, 2020. We contacted the interview participants following the second survey and 17 agreed to be interviewed again. The second set of interviews (n = 20) took place between October 23 and November 12, 2020. New interview participants were also sought for the second round of interviews, with the principle of saturation being used to determine when a satisfactory number of interviews had been conducted

[30]. After three additional interviews we determined that theme saturation had been achieved.

## Qualitative methodology

All interviews were transcribed verbatim by AJH. AJH and MM read the transcripts multiple times and initial codes were created independently following the process for qualitative content analysis outlined by Graneheim and Lundman [31] using NVivo12 to create hierarchies, merge duplicates, and examine relationships between themes. Inductive codes were developed through a close reading of the transcripts and guided by the research question and the HBM framework. Codes were reviewed for congruity and discussed between three authors (AJH, MM, AP).

## Results

In the June/July survey there were 483 respondents and, in the September/October survey there were 1269 respondents. Descriptive statistics of the demographic results are displayed in Table 1. Logistic regression analyses were run using respondents who completed all quantitative questions in the survey. This resulted in sample sizes of 433 complete cases in June/July and 1170 in September/October. Both sample sizes exceed the suggested criteria (at least 350 cases to achieve statistical power of 0.80) [32–34]. Table 2 displays the willingness of survey participants to get the COVID-19 vaccine. Table 3 displays the additional questions in the September/October survey concerning participant seasonal flu vaccine uptake and the role of their doctor/pharmacist in potential COVID-19 vaccine uptake.

### Demographics and quantitative results

**June/July survey.** In the June/July survey (n = 483) the majority of participants (77.8%) were willing to get a COVID-19 vaccine. The mean age of participants was 21.64 (Standard Deviation (SD) = 3.88) years; most participants (90.1%) were between 17 and 25 years. Respondents' perceptions of severity of COVID-19 had a mean of 5.51 (SD = 1.24).

The multinomial logistic regression analysis (Table 4) indicated that perception of the severity of COVID-19 predicted willingness to get the COVID-19 vaccine ($p < 0.001$). That is, allowing for other possible risk factors, respondents with a higher perception of the severity of COVID-19 had a greater relative chance of being willing to get COVID-19 vaccine. For each 1-point increase in perception of the severity of COVID-19 disease, participants were 2.206 times more likely to be willing to get the COVID-19 vaccine than not, controlling for all other predictor variables included in the model.

**September/October survey.** In the September/October survey (n = 1269) the majority of participants (79.6%) were willing to get a COVID-19 vaccine. The mean age of participants was 20.58 (SD = 3.31) years; most participants (93.6%) were between 17 and 25 years. Respondents' perceptions of severity of COVID-19 had a mean of 5.71 (SD = 1.30). Most participants (60.0%) reported that they do not usually receive the seasonal flu vaccine; 53.3% reported that they received a flu vaccine in 2019. The majority of participants (70.0%) indicated that they would be encouraged to get the COVID-19 vaccine if their doctor or pharmacist recommended it.

The binary logistic regression analysis (Table 5) indicates that factors predicting willingness to get the COVID-19 vaccine included being personally affected by COVID-19 ($p < 0.001$), perception of severity of COVID-19 ($p = 0.005$), and being encouraged by their doctor or pharmacist ($p < 0.001$). Controlling for other predictor variables, for students who were personally affected by COVID-19 the odds of willingness to get the vaccine are expected to

**Table 1. Demographic variables.**

| Variable | Categories | June/July n (%) | September/October n (%) |
|---|---|---|---|
| **Age** | 17–25 | 435 (90.1) | 1188 (93.6) |
| | 26–30 | 32 (6.6) | 53 (4.2) |
| | 31+ | 15 (3.1) | 26 (2.0) |
| | Did not answer | 1 (0.2) | 2 (0.2) |
| **Gender** | Male | 94 (19.5) | 343 (27.0) |
| | Female | 373 (77.2) | 897 (70.7) |
| | Gender variant/non-binary | 14 (2.9) | 26 (2.1) |
| | Prefer not to answer | 2 (0.4) | 3 (0.2) |
| **Program of Study** | Not health related | 402 (84.3) | 1136 (89.5) |
| | Health related | 75 (15.5) | 127 (10.0) |
| | Did not answer | 6 (1.2) | 6 (0.5) |
| **Household income** | Low (<$24,999 - $74,999) | 269 (55.7) | 660 (52.0) |
| | Middle ($75,000-$149,999) | 141 (29.2) | 441 (34.8) |
| | High ($150,000 <) | 51 (10.6) | 123 (9.7) |
| | Did not answer | 22 (4.6) | 45 (3.5) |
| **Location of residence** | Greater Toronto Area (GTA) | 379 (78.5) | 965 (76.0) |
| | SW Ontario other than GTA | 26 (5.4) | 68 (5.4) |
| | Northern Ontario | 8 (1.7) | 23 (1.8) |
| | Other (International) | 20 (4.1) | 110 (8.7) |
| | Western Canada | 20 (4.1) | 52 (4.1) |
| | Eastern Canada | 5 (1.0) | 11 (0.9) |
| | Quebec | 11 (2.3) | 19 (1.5) |
| | Other (Kingston) | 12 (2.5) | 20 (1.6) |
| | Did not answer | 2 (0.4) | 1 (0.1) |
| **Self-reported Anxiety** | Low (1–3) | 141 (29.2) | 319 (25.1) |
| | Moderate (4,5) | 231 (47.8) | 550 (43.3) |
| | High (6,7) | 103 (21.3) | 392 (30.9) |
| | Did not answer | 8 (1.7) | 8 (0.6) |
| **Perception of COVID-19's Severity** | Low (1–3) | 27 (5.6) | 83 (6.5) |
| | Moderate (4,5) | 194 (40.2) | 376 (29.6) |
| | High (6,7) | 259 (53.6) | 798 (62.9) |
| | Did not answer | 3 (0.6) | 12 (0.9) |
| **Affected by COVID-19** | Yes | 43 (8.9) | 149 (11.7) |
| | No | 438 (90.7) | 1117 (88.0) |
| | Did not answer | 2 (0.4) | 3 (0.2) |

decrease by 0.369, that is to decrease by 63.1%. This result should, however, be interpreted cautiously since only 11.7% of students reported being personally affected by COVID-19. Respondents with a higher perception of the severity of COVID-19 had a greater relative chance of

**Table 2. Willingness to take COVID-19 vaccine.**

| | June/July | September/October |
|---|---|---|
| **Yes** | 376 (77.8) | 1010 (79.6) |
| **No** | 56 (11.6) | 253 (19.9) |
| **Undecided** | 42 (8.7) | 0 |
| **Did not answer** | 9 (1.9) | 6 (0.5) |

**Table 3. Seasonal flu vaccine uptake and doctor/pharmacist recommendation.**

| | | September/ October |
|---|---|---|
| **Usually get flu vaccine** | Yes | 498 (39.2) |
| | No | 761 (60.0) |
| | Did not answer | 10 (0.8) |
| **Got flu vaccine last year** | Yes | 399 (31.4) |
| | No | 676 (53.3) |
| | Can't remember/not sure/no response | 194 (15.3) |
| **Would be likely to receive COVID-19 vaccine if recommended by doctor/pharmacist** | Yes | 888 (70.0) |
| | No | 176 (13.9) |
| | Undecided | 205 (16.2) |

being willing to get the vaccine. For each 1-point increase in perception of the severity of COVID-19 disease, respondents were 1.261 times more likely to be willing to get the COVID-19 vaccine than not willing to, controlling for all other predictor variables included in the model.

The contribution of doctors'/pharmacists' encouragement as a predictor variable was very strong. Respondents who indicated that they would be encouraged to get the vaccine following their doctors'/pharmacists' recommendation were approximately 76 times more likely to say they would get the vaccine than those who would not be encouraged by their doctors'/pharmacists' advice, controlling for all other predictor variables in the model.

## Qualitative results for vaccine uptake or hesitancy

**Survey free form responses and interviews.** In the June/July survey, of the participants who indicated that they would take the COVID-19 vaccine, 47 provided an open text reason (12.5%). Three participants indicated the vaccine would help their lives return to normal, 10

**Table 4. Multinomial logistic regression results.**

| Predictor | β | SE β | Wald's $\chi^2$ | p-value | Exp (β) (Odds Ratio) |
|---|---|---|---|---|---|
| Constant | -0.931 | 1.325 | -0.703 | 0.482 | 0.394 |
| Age | -0.032 | 0.041 | -0.775 | 0.439 | 0.967 |
| Gender (Male) | 0.088 | 0.402 | 0.219 | 0.827 | 1.092 |
| Program (Health Studies) | 0.608 | 0.493 | 1.236 | 0.216 | 1.838 |
| Income (Low Level)[a] | -0.703 | 0.659 | -1.067 | 0.286 | 0.495 |
| Income (Middle Level)[b] | -0.595 | 0.695 | -0.856 | 0.392 | 0.552 |
| Affected by COVID | 0.764 | 0.662 | 1.154 | 0.249 | 2.146 |
| Anxiety (Low Level)[c] | 0.167 | 0.603 | 0.277 | 0.782 | 1.182 |
| Anxiety (Moderate Level)[d] | -0.336 | 0.531 | -0.633 | 0.526 | 0.714 |
| Severity of Disease Perception[e] | 0.791 | 0.141 | 5.609 | < 0.001 | 2.206 |

a: <$24,999 - $74,999/annum

b: $75,000-$149,999/annum

c: 1–3

d: 4–5

e: $p < 0.05$.

**Table 5. Binary logistic regression results.**

| Predictor | β | SE β | Wald's χ² | p-value | Exp (β) (Odds Ratio) |
|---|---|---|---|---|---|
| Constant | -1.813 | 0.964 | -1.880 | 0.060 | 0.163 |
| Age | -0.029 | 0.029 | -0.994 | 0.320 | 0.971 |
| Gender (Male) | 0.119 | 0.239 | 0.497 | 0.619 | 1.127 |
| Program (Health Studies) | -0.271 | 0.392 | -0.690 | 0.490 | 0.763 |
| Income (Low Level)[a] | 0.338 | 0.366 | 0.921 | 0.357 | 1.402 |
| Income (Middle Level)[b] | 0.071 | 0.373 | 0.190 | 0.849 | 1.074 |
| Affected by COVID[c] | -0.996 | 0.298 | -3.341[c] | < 0.001 | 0.369 |
| Anxiety (Low Level)[d] | -0.456 | 0.299 | -1.525[d] | 0.127 | 0.634 |
| Anxiety (Moderate Level)[e] | -0.176 | 0.246 | -0.717 | 0.474 | 0.838 |
| Severity of Disease Perception[c] | 0.232 | 0.082 | 2.844 | 0.005 | 1.261 |
| Usually Get Flu Vaccine | 0.323 | 0.315 | 1.026 | 0.305 | 1.382 |
| Flu Vaccine in 2019 (Yes) | 0.650 | 0.363 | 1.789 | 0.074 | 1.916 |
| Flu Vaccine in 2019 (Not Sure) | -0.038 | 0.302 | -0.127 | 0.899 | 0.962 |
| Doctors' Recommendation (Yes)[c] | 4.332 | 0.274 | 15.797 | < 0.001 | 76.101 |

a: <$24,999 - $74,999/annum

b: $75,000-$149,999/annum

c: $p < 0.05$

d: 1–3

e: 4–5.

indicated that taking the vaccine would contribute to broader public health and safety, and two indicated they had high risk family members, thus taking the vaccine would help protect them. 23 participants indicated that they would get the vaccine but only "*[a]fter it's out for a while, in case there are side effects.*" One commented that they would get the vaccine "*RIGHT AWAY.*" Two individuals indicated they thought the vaccine should be mandatory. The remaining six participants wondered about the potential price of the vaccine. 29 participants who said no gave an open text response (51.8%). Of these, 14 participants indicated they were worried about long-term effects of the vaccine, four described their mistrust of the government, two indicated they do not take any vaccines, and one explained they had a severe phobia of needles. Three individuals said they would not take the vaccine because they had been taking sufficient precautions to avoid catching COVID-19, two thought the vaccine should be reserved for the most vulnerable individuals, and one said they would only take it if they thought they had COVID-19. Two individuals indicated their belief that vaccines should not be mandatory.

In the September/October survey we refined the question; in addition to a free text option, a list of potential reasons for the Yes/No answer was also provided (Table 6). Individuals were able to select as many reasons as they wanted. The most common reasons individuals selected to get the vaccine was to avoid catching COVID-19 (90.9%). 116 individuals who indicated that they were willing to take the vaccine chose to add a free-form response, of these 65 explained that they wanted to protect others and/or help limit the spread of COVID-19. 15 participants expressed a desire for life to return to normal, three said they wanted to travel again. 17 participants expanded upon their desire for immunity to avoid catching COVID-19; two said the vaccine should be mandatory. The final 14 individuals said they would eventually take the vaccine, but only after they felt it was safe, for example: "*I would not want the first round of the vaccine, I would want one with more long-term testing*".

**Table 6. Reasons for accepting or not accepting the vaccine (September/October survey).**

| Willing to take COVID-19 vaccine | n = 1010 |
|---|---|
| To avoid catching COVID-19 | 918 |
| To avoid illness | 667 |
| It is safe | 408 |
| Worried about becoming seriously ill | 496 |
| COVID-19 is deadlier than the seasonal flu | 499 |
| I always get the seasonal flu shot | 192 |
| I live with people who are high risk | 384 |
| I am high risk | 84 |
| I will be required to because of my job | 130 |
| Other (free form response) | 116 |
| **Not willing to take COVID-19 vaccine** | n = 253 |
| It will not work | 21 |
| Insufficient testing | 173 |
| Worried it will cause serious side effects | 168 |
| Worried it will cause bothersome side effects | 122 |
| Worried it would give me COVID-19 | 44 |
| It is not safe | 59 |
| I am not at risk of catching COVID-19 | 23 |
| I don't know where I would get it | 8 |
| Other (free form response) | 66 |

The most common reason selected for not getting the vaccine (Table 6) was a concern about insufficient testing (68.4%). 66 individuals who indicated that they would not get the vaccine entered a free form response. 28 said they wanted to wait for others to take it to understand the long-term consequences. 12 outlined their mistrust of the government and of pharmaceutical manufacturers. Five stated that COVID-19 was not serious, two said vaccines are not safe, and two said their immunocompromised state would not allow them to receive a vaccine. Three individuals indicated incorrectly that getting the vaccine could make you catch COVID-19, while one participant posited that new strains of the mutating virus could render the vaccine useless. Two participants said they would receive it only if mandated and five gave no further specific reasons. Five participants said they did not want to receive the vaccine because they were young and did not need it: *"If I catch COVID-19 at this age and health I'll be fine."* One participant noted that *"I also am in a demographic that isn't on the priority list for said vaccination"*.

Twenty individuals were interviewed in July 2020: eight male students, 10 female students, and two students who identified as gender-variant or non-binary. Individuals ranged in age from 18 to 32. The majority of interviewees (15/20) resided in the Greater Toronto Area and the majority (18/20) were not undertaking a healthcare program. All July interviewees were invited back in October and 17 chose to return. Three additional participants were interviewed in October for a total of 20. This second sample included nine males, nine females, and two individuals who identified as gender-variant or non-binary. Individuals ranged in age from 19 to 37. Similar to the July interviewees sample, the majority (15/20) resided in the GTA and the majority (18/20) were not undertaking a healthcare program. In both samples, students represented a range of self-described ethnicities and household incomes from < $24,999 to > $150,000 per annum.

In the July interviews, 16 (80.0%) of the participants said they would be willing to take the vaccine, three said no (15.0%) and one said maybe (5.0%). In the October interviews, 16 (80.0%) of the participants indicated they would take the COVID-19 vaccine when it became available, while four said they would not (20.0%). One interview participant changed their response from 'maybe' to 'yes' between the two interviews and one changed from 'yes' to 'no'. The interviews expanded upon themes displayed in Table 6, including personal timelines for getting the vaccine; safety, testing, and efficacy; and trust and reassurances. Supporting quotations for the identified themes are found in Table 7.

**Personal timelines for getting the vaccine.**   During the first round of interviews, 16 interviewees said they would get the vaccine and one said maybe. Many participants (n = 9) wanted the vaccine as soon as it became available, but there was a minority (n = 3) who wanted to wait and see how it performed.

By the second round of interviews in October, the number of participants willing to get the vaccine immediately had decreased (n = 6) and eight expressed that they preferred to wait until a large portion of the general public had received them. Participants were also quick to offer that they were not anti-vaccine, but that they had concerns about potential side effects of this particular vaccine.

**Reasons for vaccine hesitancy.**   In the first interviews three participants stated they would not take the vaccine. One had concerns about safety, the second deals with an immune condition that may preclude them from getting the vaccine, and the third has a pathological fear of needles. All three maintained their reasoning in the second round of interviews. In the second round of interviews one participant changed their response from yes to no, citing the speed of the vaccine production as their key reason for hesitancy.

**Safety and efficacy.**   All interviewees expressed concerns over safety and efficacy. Participants referred to the long development times for previous vaccines and the political pressure to have a vaccine as causes for concern. Some had very specific concerns, such as a loss of fertility. Concerns over efficacy were related to the COVID-19 vaccine being so new and related to putting faith in institutions.

**Reassurances.**   While participants were wary of issues related to safety and efficacy, they were able to identify factors that would reassure them, such as transparency during the clinical trial process, which would allow them to do their own research. Interviewees expressed trust in the scientific process and the belief that the government was invested in the health of their citizens. Participants also placed trust in the health officials who had been frequently portrayed in the news (e.g., Canada's Chief Public Health Officer Dr. Theresa Tam) and cited their backing as a reassurance.

## Discussion

In both surveys, the majority of participants stated that they were willing to get the COVID-19 vaccine, though many indicated they did not plan to receive it immediately when it became available to them. The proportion of participants willing to take the COVID-19 vaccine (June/July: 77.8% and September/October: 79.6%) was higher than Australian individuals aged 18–29 surveyed in March 2020 who said they "definitely would" get the COVID-19 vaccine (59.7%) [23]. It is worth noting, however, that 27.2% said they "probably would" get the vaccine for a combined total of 86.9% [23]. Our results were slightly lower than a survey of Italian university students, 86.1% of whom indicated they would be willing to take the COVID-19 vaccine [21], but higher than a survey of Maltese university students, of whom only 57.3% said they were 'likely' to take the vaccine [22]. Serial surveys specifically targeting university students are necessary to understand how typical these results are internationally and how they

**Table 7. Interview participant quotations.**

| Theme | | Interview quotation example |
|---|---|---|
| **Personal timelines for getting the vaccine** | As soon as available | "Yes. 100%. No doubt about it. I would line up for that vaccine. I would do it." |
| | | "I would get it not just for myself but for my family members because I know once I get the vaccine I know that my chance will go down of me even spreading it to my family members." |
| | After a personally determined period of time | "Yes, I think I would get it. But probably not the early-stage." |
| | | "I'm not going to be the first one out the door. But I am not at all opposed to vaccines." |
| | | "Like I'm not anti-vaccine. I think vaccines are great things. But it's kind of like 'oh no I need to see how it actually impacts people long term'." |
| **Reasons for vaccine hesitancy** | Concerns about speed of development | "No. I'm not anti-vaccine but I am concerned about how quickly they are being developed. . . and considering I am not going out and doing things. Like maybe if I were I would consider it more, but right now compared to potential health consequences to me just sitting at home I'm pretty sure I wouldn't get it." |
| | | "As someone who has read many articles and understands how vaccines are made, having a vaccine made in less than a year is like no. I just know. It's not happening." |
| | Participant has an auto-immune condition | "I don't know. I would have to double check with the doctors pretty thoroughly as to whether I should because some vaccines don't work for me. I haven't had a bad reaction to a vaccine before, but the potential is there." |
| | Participant has pathological fear of needles | "No. The big thing with the pandemic is I have vasovagal syncope so it's been really really bad lately because all the talk of vaccines and needles online. . .personally I can't process the thought of a vaccine." |
| **Safety and efficacy** | Participant notes longer development times for previous vaccines | "Usually it takes like 10 years, and they are constantly pushing it forward, pushing it forward." |
| | Fear of loss of fertility | "I want to make sure that it's safe and that there is no way that it would impact my fertility. So yes, but I want to make sure that it's safe and it's tested." |
| | Effectiveness and potential side effects | "I would get it and I would encourage everyone I know to get it depending on how effective it is. Well, I don't know. I have heard some news say that there is no proof that antibodies develop." |
| | | "We do not want people to be taking vaccines that are not effective, or taking vaccines that are effective but that have some really drastic side effects, particularly with people who are already sceptical of vaccines, we would not want to give them a real reason to fret about it." |
| | Putting faith in institutions | "With the COVID vaccine, the efficacy and safety are, it's a leap of faith as it were, it's faith essentially. You are placing your faith in scientists and the government." |
| **Reassurances** | Lack of vaccine injuries | "No one died. That's what comes down to it. If it was a month or so and people were producing antibodies I would be the first one to line up the next morning." |
| | Transparency during clinical trial process | "I think they are probably going to come out with those papers when it's all said and done, but I think it would be beneficial if they did that in a step-by-step kind of thing. One research step, one testing step, and then like, 'let's get out a research paper'." |
| | | "When it comes down to it, there needs to be so much transparency. Like, I want to know what brand of pipette they used." |
| | Trust in government | "I would like to say that here in Canada we can trust our scientific community and our medical community and we can trust that if they say that you can take this vaccine and it won't cause any harm, I like to trust that it is being based on nothing but empiricism." |
| | | "I would be informed about where it's coming from and who, like what form of it, but I trust Ontario regulators, Canadian regulators. . .I will just do my due diligence." |
| | Trust in public health officials | "I would wait for those healthcare people like Fauci and the guy from Maine. And that one, I remember her last name is Tran [Canada's Chief Public Health Officer Dr. Theresa Tam], you know the one from Canada, that person, I don't follow her, but I retweet her. People like that I would follow people like that I would look to, to make sure it's a real vaccine and not a rushed one for political purposes." |

may vary depending on the relative regional severity of COVID outbreaks. Further, recognizing the potential delay between vaccine introduction and actual administration to university-aged students should be taken into account.

Consistent with the HBM, an individual's personal perception of the severity of COVID-19 was a clear predictor of willingness to get the COVID-19 vaccine in both surveys. This result is

similar to Reiter and colleagues' survey of adults concerning COVID-19 in the United States [35] and Wong et al.'s (2020) [19] survey in Hong Kong. Setbon and Raude, in reference to uptake of the H1N1 vaccine, noted it was necessary that the risk of swine flu be perceived to be higher than the risk of the seasonal flu to encourage vaccine uptake [36]. 49.4% of individuals who indicated they would take the vaccine noted that "COVID-19 is deadlier than the seasonal flu" as a reason influencing their decision (Table 6). These results suggest that not only is public health education focusing on COVID-19 severity essential as a cue to action, but that education should include specific information on the immediate and emerging long-term health risks associated with catching COVID-19.

There is a clear relationship between an individual getting their doctors'/pharmacists' recommendation and willingness to receive the vaccine. This suggests that healthcare workers have the ability to act as modifying variables within the HBM and can influence vaccine uptake. Healthcare worker input has previously been linked to willingness to take the flu vaccine [37] and was found to be a key predictor in a survey of adults in the United States regarding their uptake of the COVID-19 vaccine [35]. In this study, participants' previous uptake of the flu vaccine was not associated with willingness to get the COVID-19 vaccine. This result differs from previous studies investigating H1N1 vaccine uptake, which found that adult individuals' past vaccinations were a predictor [36–38]. Statistics Canada reported that seasonal flu vaccination coverage for all adults (18+) in 2019–2020 was 41.8%, though coverage for adults 18–64 with no chronic medical conditions was only 30% [39]. 31.4% of the September/October survey participants reported getting the flu vaccine in 2019–2020, demonstrating that this research sample resonates with larger national results. Previous behaviour was not a predictor in this research, indicating that public outreach should focus on alternative means of encouraging vaccine uptake.

The qualitative results drawn from free-form survey responses and interviews revealed a nuanced understanding of both positive and negative views of the vaccine and the intention to receive it. Participants were heavily motivated by self-protection, which was seen as potential benefit to receiving the vaccine. Self-protection as a key motivating factor was found in another COVID-19 mixed methods study in England [40] and the majority of survey participants in September/October (90.9%) indicated that avoiding catching COVID-19 was a key reason for getting the vaccine. Promoting altruism has been suggested as a means of increasing vaccine uptake and shifting vaccine decisions towards questions of community good [41]. Altruism was mentioned by some participants and encouraging this perspective through public health education could potentially increase uptake for participants who are not motivated by self-protection.

Despite the majority of participants indicating they would be willing to receive the COVID-19 vaccine (Table 2), there were vaccine hesitancy concerns raised as perceived barriers within the HBM. Vaccine safety relating to the speed of development was a key theme highlighted, which resonates with previous studies concerning decision making surrounding vaccines [42] and recent studies on COVID-19 vaccines [40, 43]. Research into parental attitudes towards new vaccines found that parents tended to express concern about newly available vaccines and their judgement of the severity of a potential disease or illness influenced their willingness to allow their child to be vaccinated [44, 45]. One interviewee changed their mind from 'yes' to 'no' within four months because of their perception that the vaccine was being developed and approved too quickly. As the survey results indicate there is a strong sense of trust in healthcare providers, it will be important for frontline healthcare workers to have information specifically addressing the speed of vaccine production and approval to help counteract these barriers with specific cues to action. Mercadante & Law's [20] study of American patient intentions of

taking the COVID-19 vaccine using the HBM indicated that public health campaign cues to action have an impact upon vaccine intention.

Understanding university students' perceptions of the COVID-19 vaccine is important, because as noted by one survey participant, most are not "*in a demographic. . .on the priority list for said vaccination.*" The individuals surveyed (average age: 21.64 and 20.58, June/July and September/October surveys respectively), unless dealing with co-morbid conditions, are of the lowest priority for the current vaccination program [24]. While these individuals wait their turn, it will be important to provide tailored encouragement for vaccine uptake. The HBM notes that clinical recommendations are more likely to be followed if the individual perceives that the benefits of the recommendation outweigh the barriers [46]. Clear and consistent communication from clinical providers, as well as public health campaigns that acknowledge potential barriers to vaccination, are necessary.

## Strengths and limitations

The survey participants are a convenience sample with a majority of female-identifying participants. Study participants were drawn from one university in a major urban centre in Ontario and thus may not be suitable for generalizations to be drawn about other university contexts of broader populations. Our recruitment strategies resulted in a smaller sample size in the June/July period, likely due to a smaller number of students taking summer session courses. The second sample size was larger, as more students were taking classes and interacting with social media related to the university. As our primary sample was young adults in university, we did not control for level of education as a predictor. Previous studies have found a relationship between level of education and intent to vaccinate [47].

Despite these limitations, this research provides data on an important subsection of the population. This research used a multi-methods approach, which exposes further nuance underlying the quantitative results. This research included two rounds of survey and interview responses, responding to a need for longitudinal and repeated studies to capture emerging data [40]. We asked survey participants to express their perceptions about vaccination at a time when vaccines were in early testing and/or clinical trial periods. It is possible that their perceptions will evolve as further data become available.

## Conclusions

Vaccine uptake is a critical issue. Using the HBM as a framework, our serial surveys indicate that both early in the pandemic, before vaccinations were nearing widespread approval or availability, and during clinical trials, nearly 80.0% of university students intended to receive the COVID-19 vaccine, though projected personal timelines of uptake varied. Despite the majority indicating that they were willing to get the vaccine, students still expressed hesitation concerning barriers, such as the safety of the vaccines related to the speed at which they were being produced and tested. Higher individually perceived severity of COVID-19 was predictive of vaccine uptake intention in both surveys. In the second survey, a doctor's/pharmacist's recommendation (cue to action) to take the COVID-19 was predictive of participants' willingness to get the vaccine. Transparency from public health and governmental bodies is critical to ensure public trust and uptake as university students wait their turn to be eligible for the vaccine. The trust in healthcare providers expressed by the students is a key result; frontline healthcare practitioners should aim to provide clear and consistent communication about vaccine safety to encourage vaccine uptake among university students.

## Supporting information

**S1 File. Survey and interview details.** June/July 2020 and September/October 2020 survey questions, semi-structured interview questions.
(DOCX)

## Author Contributions

**Conceptualization:** Madeleine Mant, Andrew Prine, Alyson Jaagumägi Holland.

**Data curation:** Madeleine Mant, Alyson Jaagumägi Holland.

**Formal analysis:** Madeleine Mant, Asal Aslemand, Andrew Prine, Alyson Jaagumägi Holland.

**Funding acquisition:** Madeleine Mant.

**Investigation:** Madeleine Mant, Asal Aslemand, Andrew Prine, Alyson Jaagumägi Holland.

**Methodology:** Madeleine Mant, Asal Aslemand, Andrew Prine, Alyson Jaagumägi Holland.

**Project administration:** Madeleine Mant, Alyson Jaagumägi Holland.

**Resources:** Madeleine Mant, Asal Aslemand.

**Software:** Asal Aslemand.

**Supervision:** Madeleine Mant.

**Writing – original draft:** Madeleine Mant, Alyson Jaagumägi Holland.

**Writing – review & editing:** Madeleine Mant, Asal Aslemand, Andrew Prine, Alyson Jaagumägi Holland.

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
