## [Decision Letter · Decision Letter 0]

15 Jun 2021

PONE-D-21-08664

University students’ perspectives and hesitancy regarding the COVID-19 vaccine: a multi-methods study

PLOS ONE

Dear Dr. Mant,

Thank you for submitting your manuscript to PLOS ONE. After careful consideration, we feel that it has merit but does not fully meet PLOS ONE’s publication criteria as it currently stands. Therefore, we invite you to submit a revised version of the manuscript that addresses the points raised during the review process.

Please edit your presentation of results as recommended by reviewers.

Please attend to the comment on cost of vaccine per reviewer #2.

We look forward to receiving your revised manuscript.

Kind regards,

James Curtis West, M.D.

Academic Editor

PLOS ONE

Journal Requirements:

2. Please include a copy of the interview guide used in the study, in both the original language and English, as Supporting Information, or include a citation if it has been published previously.

Additional Editor Comments (if provided):

Thank you for your interesting submission. Your investigation of vaccine acceptance and hesitance among university students is both timely and well executed. Please consider the following suggestions to improve.

1. Results. Please edit down your presentation of the interview data. I would recommend removing table 7 in favor of just a paragraph of general descriptives of the sample. Also, it is not necessary to identify each participant alongside the quotes. i would strongly urge you to consider placing the quotes into a table or text box organized by the themes identified in your text.

2. Results. Table 1, please clarify your label "severity." I believe this is referring to respondents' perception of the severity of COVID, but coming right after "anxiety" it gets confusing as a single word label.

3. Results. line 239. Your description of students not personally affected being 0.369 times less likely is the opposite of how it is presented in the table and the use of a fractional reduced likelihood is awkwardly worded.

Reviewers' comments:

Reviewer's Responses to Questions

**Comments to the Author**

1. Is the manuscript technically sound, and do the data support the conclusions?

Reviewer #1: Yes

Reviewer #2: Yes

2. Has the statistical analysis been performed appropriately and rigorously? 

Reviewer #1: Yes

Reviewer #2: Yes

3. Have the authors made all data underlying the findings in their manuscript fully available?

Reviewer #1: Yes

Reviewer #2: No

4. Is the manuscript presented in an intelligible fashion and written in standard English?

Reviewer #1: Yes

Reviewer #2: Yes

5. Review Comments to the Author

Reviewer #1: This study investigated the university students' willingness to receive a COVID-19 vaccine using a mixed method and then identified the concerns on efficacy and safety of COVID-19 vaccines, which is similar to other publications elsewhere.

1, The presentation of Table 4 and 5 should be revised, as usual appearance in a scientific paper.

2, I suggest the qualitative results should be revised and refined.

Reviewer #2: In my opinion, this is a very interesting study assessing the perspectives and hesitancy of University students regarding the COVID-19 vaccines, as studies on this particular population regarding COVID-19 are still limited. The aim of the study, methods used, and results were presented clearly. However, there are some suggestions I'd like to give to enrich the manuscript.

1. The title of this study is "University students' perspectives and hesitancy regarding the COVID-19 vaccine: a multi-methods study," however the elaboration on hesitancy part is insufficient. Further elaboration on the issue in the study, particularly in the discussion and conclusion, is necessary to better fit the title.

2. In regards to the likeliness of university students to get COVID-19 vaccine once it's available, the authors did not include price and regulation as predictors or factors that might influence decision to get vaccinated. In fact, these two factors have been shown to be associated with vaccination uptake in previous investigations. Wether or not the COVID-19 vaccine will be available for free, and if it will be made compulsory by the government might influence the decision of this particular population to get themselves vaccinated. In my opinion, it would be great if the authors could explain why they did not include the two factors in the introduction part (it was explained briefly in the methods section, but no justification of why the two items were not included in the survey).

All in all, I think the paper was well designed and suit to be published.

6. PLOS authors have the option to publish the peer review history of their article (what does this mean?). If published, this will include your full peer review and any attached files.

Reviewer #1: **Yes: **Yihan Lu

Reviewer #2: No

---

## [Author Response · Author response to Decision Letter 0]

25 Jun 2021

June 25, 2021

Response to Reviewers

Thank you to the Reviewers and the Academic Editor for their reviews and comments. We appreciate the time spent assessing this manuscript and hope our revisions have fully addressed the identified points. The response/actions to the comments are noted below. All four authors have approved the following changes to the manuscript.

Please include a copy of the interview guide used in the study, in both the original language and English, as Supporting Information, or include a citation if it has been published previously.

Response: The June and September survey questions and interview guides are included as Supporting Information.

Editor Comments

1. Results. Please edit down your presentation of the interview data. I would recommend removing table 7 in favor of just a paragraph of general descriptives of the sample. Also, it is not necessary to identify each participant alongside the quotes. i would strongly urge you to consider placing the quotes into a table or text box organized by the themes identified in your text.

Response: Thank you for these suggestions. All references in parentheses to specific participant numbers have been removed from the text. Table 7 has been removed and replaced with a paragraph summarizing the interview participant sample demographics. All interviewee quotations have been removed from the body text and organized into a table (new Table 7) based upon the themes identified in the Results. 

2. Results. Table 1, please clarify your label "severity." I believe this is referring to respondents' perception of the severity of COVID, but coming right after "anxiety" it gets confusing as a single word label.

Response: Table 1 has been adjusted so that Anxiety now reads “Self-reported Anxiety” and Severity now reads “Perception of COVID-19’s Severity” to clarify that these are separate sets of results. 

3. Results. line 239. Your description of students not personally affected being 0.369 times less likely is the opposite of how it is presented in the table and the use of a fractional reduced likelihood is awkwardly worded.

Response: Thank you for catching this. The text has been revised to accurately reflect what is presented in Table 5.

Reviewer 1

1, The presentation of Table 4 and 5 should be revised, as usual appearance in a scientific paper.

Response: Tables 4 and 5 have been revised to align with APA formatting for reporting logistic regression. 

2, I suggest the qualitative results should be revised and refined.

Response: Following the comments provided by the Academic Editor, and as detailed above, the references to specific participants have been removed from the main text, Table 7 was removed and replaced with a paragraph summarizing the interview participant sample demographics, and all interviewee quotations have been removed from the text and placed into a new Table 7, organized by the themes identified in the Results.

Reviewer 2

1. The title of this study is "University students' perspectives and hesitancy regarding the COVID-19 vaccine: a multi-methods study," however the elaboration on hesitancy part is insufficient. Further elaboration on the issue in the study, particularly in the discussion and conclusion, is necessary to better fit the title.

Response: Thank you for this point. We have adjusted the title to “University students’ perspectives, planned uptake, and hesitancy regarding the COVID-19 vaccine: a multi-methods study” in order to better reflect the aims and content of the paper. The subsection “Qualitative results for vaccine uptake or resistance” has been revised to “Qualitative results for vaccine uptake or hesitancy” and the subsection “Reasons for not receiving the vaccine” was revised to “Reasons for vaccine hesitancy.” The section discussing reasons for vaccine hesitancy has been revised to explicitly state “there were vaccine hesitancy concerns raised as perceived barriers within the HBM.” The speed of development was the key fear expressed by survey participants, which is discussed and placed into the wider context of the limited existing literature. 

2. In regards to the likeliness of university students to get COVID-19 vaccine once it's available, the authors did not include price and regulation as predictors or factors that might influence decision to get vaccinated. In fact, these two factors have been shown to be associated with vaccination uptake in previous investigations. Wether or not the COVID-19 vaccine will be available for free, and if it will be made compulsory by the government might influence the decision of this particular population to get themselves vaccinated. In my opinion, it would be great if the authors could explain why they did not include the two factors in the introduction part (it was explained briefly in the methods section, but no justification of why the two items were not included in the survey).

Response: This is an important point that is dependent upon the national context in which this research was conducted. Regarding vaccine cost, routine vaccinations for babies, children, and adults are offered free of charge across Canada. Seasonal flu shots are free and the mass immunization program for the H1N1 pandemic in 2009 provided free vaccines for all Canadians who wanted a vaccine. We did not consider cost as a driving factor because the universal health care system in Canada covers the cost of both routine and pandemic vaccines. We also did not consider government mandating of the vaccine because there are no mandated vaccines in Canada (though two provinces have specific rules regarding vaccines for school-aged children to register in public schools, which have been noted in the text. They do, however, have medical and philosophical exemptions.) Survey participants spontaneously offered thoughts regarding making the vaccine mandatory, which has been noted in the results. We added an explanation (in the Methods section, subsection on Survey questions) as to why cost and mandatory vaccines were not considered as potential drivers of the survey participants’ answers. Three new citations have been added to the manuscript as part of this revision (#27-29).

---

## [Editor Report · Decision Letter 1]

19 Jul 2021

University students’ perspectives, planned uptake, and hesitancy regarding the COVID-19 vaccine: a multi-methods study

PONE-D-21-08664R1

Dear Dr. Mant,

We’re pleased to inform you that your manuscript has been judged scientifically suitable for publication and will be formally accepted for publication once it meets all outstanding technical requirements.

Kind regards,

James Curtis West, M.D.

Academic Editor

PLOS ONE
---

## [Editor Report · Acceptance letter]

22 Jul 2021

PONE-D-21-08664R1 

University students’ perspectives, planned uptake, and hesitancy regarding the COVID-19 vaccine: a multi-methods study 

Dear Dr. Mant:

I'm pleased to inform you that your manuscript has been deemed suitable for publication in PLOS ONE. Congratulations! Your manuscript is now with our production department. 

Kind regards, 

on behalf of

Dr. James Curtis West 

Academic Editor

PLOS ONE